# Influence of Extrusion Temperature on Properties of Graphene Oxide-Carbon Fiber/Epoxy Composite Prepared by Vacuum Infiltration Hot-Press-Forming Experimental System

**DOI:** 10.3390/nano12213839

**Published:** 2022-10-30

**Authors:** Yuqin Ma, Fei Li, Wei Xu, Yuyang Zhang, Yi Xu, Haiyin Guo, Yatao Li

**Affiliations:** 1State Key Laboratory of Public Big Data, Guizhou University, Guiyang 550025, China; 2Key Laboratory of Road Construction Technology and Equipment of MOE, School of Construction Machinery, Chang’an University, Xi’an 710064, China; 3School of Mechano-Electronic Engineering, Xidian University, Xi’an 710071, China; 4School of Aeronautical Engineering, Shaanxi Polytechnic Institute, Xianyang 712000, China

**Keywords:** VIHPS, graphene oxide, carbon fiber composites, extrusion temperature, microstructure, bending properties

## Abstract

Graphene oxide-carbon fiber/epoxy (GO-CF/EP) composites with extrusion temperatures of 30, 40, 50, 60 and 70 °C were prepared by a vacuum infiltration hot-press-forming experimental system (VIHPS). The effects of extrusion temperature on the microstructure, fracture mechanism and mechanical properties of GO-CF/EP composites were investigated. It was found that the best mechanical property of composites and infiltration effect of the matrix in the fiber gap were obtained at the temperature of 50 °C, and the bending strength of the composite reached 728 MPa. The fiber was pulled out and broken under the wrapping of the matrix. The matrix viscosity was high, and the fluidity was poor when the extrusion temperature was low. The poor infiltration of the matrix resulted in many fibers failing to bond together, resulting in the disorderly breakage of fiber bundles. Under the condition of higher temperature, the flow speed of the matrix could be improved. However, part of the matrix was extruded during the extrusion process, and cracks and other defects occurred during the loading, which caused the brittle fracture of the specimen.

## 1. Introduction

Graphene oxide-carbon fiber/epoxy (GO-CF/EP) composite is a new type of composite with nano GO added into carbon fiber-reinforced polymer (CFRP) composite. GO is widely used to enhance the performance of composites due to its excellent mechanical, electrochemical and thermal properties. The composites with GO significantly improve the mechanical properties and apparent advantages in optical, thermal and shielding properties [1,2,3,4,5].

The properties of CFRP composites have been widely studied, but the low surface energy and surface hydrophobicity of CF makes the interface adhesion between CF and the matrix poor. Due to the poor interface performance, the effective transfer of stress between matrix and fiber is hindered [6,7]. Therefore, nanomaterials such as GO have become ideal reinforcing fillers to improve the surface properties of CF and enhance the interfacial bonding strength between CF and the matrix. GO is a two-dimensional intermediate formed by the exfoliation of graphene during oxidation [8]. The study of GO-reinforced polymer matrix was earlier than that of GO-reinforced CFRP as a nanofiller. Xiao and others [9] proposed a new method to synthesize polyaniline-intercalated GO nanocomposites. The experimental results showed that the compound had a higher crystallinity regularity. Szabo et al. [10] deposited thin GO/cationic polymer (poly (diallyldimethylammoniumchloride, PDDA) multi-layer films on a glass substrate through wet chemical self-assembly. Through experimental research, they found that the multi-layer combination of GO and PDDA reduced to a non-conductive turbine-layer carbon nanofilm on the glass. Matsuo et al. [11] prepared poly (epoxy oxide)-intercalated graphite oxides (PEO-intercalated GOs) with different c-axis repeat distances and characterized them in detail. The decomposition temperature of GO was different, and GO decomposed at higher temperatures was accompanied by the destruction of the carbon skeleton. However, the strong hydrogen bond of oxygen-containing groups between GO layers and the dipole interaction hinders the dispersion and exfoliation of GO in polymer aggregates [12]. Therefore, researchers mainly use mechanical, ultrasonic, and microwave radiation to obtain the matrix solution containing GO. It is dispersed in organic solvent and can be peeled into a single-layer structure after ultrasonic treatment, which is conducive to the uniform dispersion of GO in the solvent. The uniform dispersion ability in the matrix is significantly improved [13]. In addition, the chemical functional groups on the GO surface react with the groups on the fiber surface, which promotes wettability between the matrix and CF, and enhances the interface adhesion between them, so the composites have better interface performance [14].

The preparation methods of CF-reinforced composites mainly include hot pressing, molding, extrusion, winding, and so on. Guo [15] prepared CF shape-memory composites by hot pressing and studied the influence of CF content on the mechanical and shape-memory properties of shape-memory composites. Li [16] prepared CF-reinforced shape-memory composites through the molding process and studied the influence of fiber content on the shape-memory bending recovery performance of the composites. Boss and Ganesh [17] prepared CF fiber and Kevlar fiber-reinforced gradient composite rods by combining fabric performing and pultrusion processes. They proposed the possibility of changing the mechanical properties of the composite rods by their length. Zhang et al. [18] prepared CF-reinforced epoxy composites by winding and laying up fiber prepregs. The tensile and inter-laminar shear properties of the composites were studied. If the above processes are used to prepare GO-CF/EP composites, there will be problems such as long preparation time, low preparation efficiency, inability to achieve the complete mixing of GO and EP, and complex operation. It is difficult to determine the impact of the preparation process on the properties of the composites. Therefore, this paper combined the wet transfer method, electromagnetic stirring, and ultrasonic dispersion method to prepare the matrix solution containing GO and used the vacuum infiltration hot-press-forming experimental system (VIHPS) to prepare GO-CF/EP composites. Extrusion molding can timely exhaust the gas among the fibers and make the mixed solution fully infiltrate the fiber gap to reduce the holes, lamination and other defects of the composites. It can also improve the density and bearing capacity of the composites [19]. In order to obtain composites with sufficient microstructure infiltration and ideal mechanical properties, the process parameters should be strictly controlled in the forming process. The extrusion temperature is an essential parameter in the hot-pressing process. If the extrusion temperature is too low, the flow viscosity of the matrix will be affected, and the effect of the matrix infiltration will be poor. If the extrusion temperature is too high, the matrix is extruded under the effect of extrusion pressure. The matrix cannot completely penetrate into the fiber gap, affecting the mechanical properties of the composite [20]. Therefore, choosing the appropriate extrusion temperature is the key to preparing GO-CF/EP composites with excellent mechanical properties by using VIHPS.

At present, relevant researchers have conducted many experiments and studies on the influence of extrusion temperature on composites. Fard et al. [21] studied the effects of extrusion temperature on the microstructure and porosity of A356-silicon carbide particles (SiCp) composites. It was found that the grain refinement of SiCp decreased first and then increased with the increase in extrusion temperature, while the porosity of the composites decreased. Wang et al. [22] obtained the variation of the tensile strength of SiCp/Mg-Zn-Ca matrix (Mg-Zn-Ca) composites with different extrusion temperatures. The results showed that the grain refinement of SiCp varies with extrusion temperature. Shao et al. [23] prepared graphene nanosheet-reinforced Al6061 composites by pressure infiltration and studied the influence of extrusion temperature on the composites. It turned out that the deformation behavior and dispersion of graphene were different due to different extrusion temperatures. Ma et al. [24] prepared two-dimensional carbon fiber-reinforced aluminum matrix (2D-Cf/Al) composites by the liquid extrusion and infiltration method and studied the effects of extrusion temperature on the microstructure and mechanical properties of composites.

The above studies have discussed the influence of extrusion temperature on metal matrix composites, but there is a lack of polymer matrix composites. When VIHPS is used to prepare GO-CF/EP composites, there are some problems, such as the unclear reinforcement effect of GO on the matrix and CF; different infiltration effects of the matrix between fiber bundles and layers, resulting in uneven infiltration; and the unclear influence of changes in the process parameters on the properties of composites. The influence of VIHPS on the properties of GO-CF/EP composites needs to be solved urgently.

In order to explore the effect of extrusion temperature on the infiltration structure and mechanical properties of GO-CF/EP composites, the micro-nano scale hybrid reinforced composites were prepared by VIHPS in this study. Combined with the microstructure characterization, mechanical property test and fracture morphology observation at different extrusion temperatures, the influence of extrusion temperature on GO-CF/EP composites was analyzed and summarized. The experimental results will lay a theoretical and experimental foundation for later preparation and further research of GO-CF/EP composites.

## 2. Materials and Methods

### 2.1. Experimental Materials

The GO used in this study is prepared by Shenzhen Suiheng Technology Co., Ltd. (Shenzhen, China) using an improved Hummers process, with a lamellar diameter of 0.2–10 μm, the thickness of 1 nm, a layer number of carbon atoms connected with oxygen-containing functional groups of GO powder of 1–2 and a purity of 99%. The epoxy and curing agents are E-51 epoxy resin and phenolic amine T31 curing agents produced by Nantong Xingchen Synthetic Composite Co., Ltd. (Nantong, China) T700 unidirectional CF cloth is produced by Toray Co., Ltd (Shanghai, China). Its tensile strength is 3920 MPa, and the elongation is 1.71%. The ethanol solution with 99.7% concentration is produced by Jiangsu Qiangsheng Functional Chemistry Co., Ltd (Changshu, China).

### 2.2. Vacuum Infiltration Hot-Press-Forming Process

VIHPS comprises four parts: the fiber preform module, vacuum heating infiltration module, hot-press curing molding module and data acquisition and control module [25]. The fiber preform module mainly includes a fiber cloth cutting device, stirring device and ultrasonic dispersion device. The module can preliminarily impregnate the solution into the fiber cloth lamination and fiber gap to obtain the CF preform. The vacuum-heating infiltration module mainly includes a heating drying oven, vacuum pump and so on. The purpose of heating and drying is to promote the fluidity of the matrix solution. The vacuum environment can cause the composite to exhaust excess gas, which is conducive to the complete immersion of the solution in the fiber. After vacuum infiltration, the surface of the composite is solidified, but the interior is still in an incomplete solidification state. It needs to be solidified entirely by hot extrusion at a specific temperature. The final processing of composites is carried out by a hot-press curing molding module, which mainly includes a heating mold, hydraulic press and other devices. The composite can have a certain thickness and obtain the required mechanical properties after hot pressing. In the whole preparation system, temperature sensors, pressure sensors, digital vacuum meters and other controllers are used to convert non-electrical signals into electrical signals and input them into computers. The computer controls all parameters in real time. When the parameters are abnormal, they can be adjusted in time to meet the production requirements. The composition of the whole system is shown in Figure 1.

#### 2.2.1. Preparation of Curing Mixed Solution

Due to the two-dimensional surface structure, high surface energy, strong hydrogen bond and dipole interaction of the interlayer oxygen-containing groups of GO, the dispersion and shedding behavior of GO in the resin are hindered [12]. In order to improve the aggregation and precipitation of GO in the matrix, the combination of electromagnetic stirring and ultrasonic dispersion is used to prepare the curing mixed solution. The specific preparation process is as follows. GO powder was mixed with absolute ethanol solution and initially stirred manually. The mixed solution was transferred to an electromagnetic stirring device and stirred at 500 rpm for 20 min. The solution was ultrasonically dispersed at 40 kHz and 200 W for 60 min to obtain a GO suspension. The EP was added to the suspension, and the viscosity of the EP was large at normal temperature, which is not conducive to the uniform stirring of the matrix. Therefore, the solution was placed on an electromagnetic stirring device at 50 °C and stirred at 500 rpm for 20 min. Afterward, the solution was dispersed on an ultrasonic dispersion device for 60 min. The uniform mixing process of the solution was completed. Finally, the solution was vacuum treated for 24 h under a vacuum degree of −0.09 MPa. After the vacuum treatment, the curing agent was added to form a curing mixed solution containing GO. The preparation process is shown in Figure 2.

#### 2.2.2. Preparation Process of GO-CF/EP Composites

The preparation process of GO-CF/EP composites strictly follows the VIHPS. (1) T700 CF cloths with a size of 100 mm × 95 mm were cut, the cured mixed solution was applied on both sides of each fiber cloth, and CF cloths were stacked in turn and placed for preliminary extrusion to discharge some internal bubbles. (2) The preform was placed in the molding machine, which cured it at room temperature for 2 h. The extrusion pressure was set to 0.2 MPa. In this process, the thickness of the test piece was set to 2 mm. After that, the preform was transferred to a vacuum drying oven with a vacuum of −0.09 MPa and a temperature of 50–70 °C for 1 h. The high temperature is conducive to improving the fluidity of the solution, thoroughly infiltrating the matrix, and the vacuum environment can effectively prevent the generation of pores, bubbles and other defects. (3) The preforms were taken out and placed in the pre-heated hot pressing die to ensure that the pressure was 0.5 MPa. The extrusion temperature was controlled to be 30, 40, 50, 60 and 70 °C in turn, and the pressure was maintained for 5 min. (4) The composites were removed after the mold cooled to room temperature when the hot-pressing process was stopped. The process is shown in Figure 3.

### 2.3. Testing and Characterization

The bending properties of GO-CF/EP composites at different extrusion temperatures were tested by a three-point bending test according to GB/T3356-2014. The bending strength ($\sigma_{f}$) and bending modulus ($E_{k}$) of the specimen were calculated from Equations (1)–(3). The three-point bending test instrument was the DNS 100 electronic universal testing machine of Changchun Machinery Institute, which was applied to the bending, tensile, compression, and other mechanical property tests. It can calculate the maximum test force, bending strength, tensile strength, compressive strength, elastic modulus, and other parameters. The maximum test force can provide 100 kN, and the measurement accuracy is ±0.5%. The loading head and bearing radius were 5 mm, the span thickness ratio was 32:1, the loading speed was 5 mm/min, and the specimen size was 80 mm × 12.5 mm × 2 mm. For each group of extrusion temperature, 5 test pieces were tested. The size of the composite prepared in 2.2.2 was 100 mm × 95 mm × 2 mm. Six specimens of size 80 mm × 12.5 mm × 2 mm were cut out, five of which were used for the bending performance test and one for the microstructure test. The loading diagram is shown in Figure 4.
(1)$$\sigma_{f}=\frac{3P_{max}L}{2\omega h^{2}}$$
(2)$$\varepsilon_{f}=\frac{6\delta h}{L^{2}}$$
(3)$$E_{k}=\frac{\Delta \sigma }{\Delta \varepsilon }$$
where $\sigma_{f}$ is the bending strength of the specimen (MPa), $P_{max}$ is the maximum load of the sample (N), L is the span (mm), h is the thickness of the sample (mm), ω is the specimen width (mm), $\varepsilon_{f}$ is the failure strain (mm/mm), δ is the deflection of the specimen in the middle of the span (mm), $E_{k}$ is the bending modulus of the specimen (GPa), Δσ is the difference of bending stress between two selected strain points (MPa), and Δε is the difference of strain between two selected strain points (mm/mm).

The internal microstructure and bending fracture section morphology of the composites after impregnation was observed by the JEOL JSM-6390A scanning electron microscope, which is a high vacuum analytical scanning electron microscope purchased from Japan Electronics Co., Ltd (Beijing, China). The size of the specimen for microstructure testing is 10 mm × 12.5 mm × 2 mm, the specimen for fracture morphology test needs to be cut after a three-point bending test of the specimen, and the size is close to 10 mm × 12.5 mm × 2 mm. Three specimens are tested at the same extrusion temperature.

## 3. Results and Discussion

### 3.1. Characterization of GO

Some chemical links between GO, EP and CF can effectively enhance the interaction [26]. One study showed that the dispersion uniformity of GO in the matrix is improved when pressing force is applied during the infiltration process, and its distribution trend is closely related to the extrusion direction and matrix fluidity [23]. As shown in Figure 5a, the surface of GO is uneven and has a large number of folds. The fold structure endows GO with a larger surface area so that it can adhere to the surface of the fiber or resin matrix. The distribution of GO on resin or fiber is shown in Figure 5b. Figure 6 shows the schematic diagram of GO playing the role of mechanical locking in GO-CF/EP composites. The contact area between the CF and matrix can be increased and a good interface bonding can be formed by adding GO to the composite. In addition, the unique structure of GO causes the mechanical locking between EP and CF, thus, enhancing the interface bonding strength of the composite. This is helpful to the stress transfer of the specimen in the loading process. The maximum stress is often near the reinforcement and matrix interface when the composite is subjected to an external force. Ye et al. [27] also obtained a similar conclusion. Therefore, the mechanical lock and structure formed by the addition of GO can increase the maximum stress of the composite to a certain extent. The presence of GO can also reduce the concentration of internal stress, play a role in toughening, and disperse the coarse cracks concentrated in the layer into fine cracks [28].

### 3.2. Infiltration Structure of GO-CF/EP Composites

The microstructures of the composites prepared at different extrusion temperatures are different. As shown in Figure 7a,b, when the extrusion temperature is low, there is only a small amount of matrix distribution in fiber bundles. At this time, the matrix has a large viscosity, which hinders its infiltration behavior in fiber bundles, so the matrix is mainly distributed on the upper and lower surfaces of the specimen or between the fiber cloth layers. The matrix content between fiber bundles gradually increases with the increase in extrusion temperature. When the temperature reaches 50 °C (Figure 7c), the matrix containing GO is evenly distributed, and fiber bundles form a continuous infiltration structure under the adhesion of the matrix, which is conducive to stress transmission and crack dispersion. When the extrusion temperature reaches 60 and 70 °C (Figure 7d,e), the matrix has a good flow performance and can quickly fill the gaps in the specimen. However, with the extension of extrusion time, part of the matrix is extruded under the effect of extrusion pressure, resulting in two kinds of infiltration of the composite: matrix aggregation areas and matrix-free areas, resulting in uneven infiltration. In the process of loading, the uneven structure of the infiltration promotes the specimen to have different mechanical behaviors, reducing its bearing capacity.

### 3.3. Bending Properties of GO-CF/EP Composites

#### 3.3.1. Bending Strength and Modulus of GO-CF/EP Composites

As shown in Figure 8a, the load increases with displacement at the beginning of the loading stage. With the increase in displacement and load, when the load exceeds the limit load of the composite, cracks begin to appear on the lower end of the specimen, and the composite is destroyed. When the extrusion temperature is 50 °C, the maximum bending load of the composite is 455 N. However, the maximum load that the specimen can bear under other extrusion temperatures is below 400 N. When the temperature rises to 70 °C, the maximum load is only 251.8 N.

Figure 8b shows the bending strength and the bending modulus of GO-CF/EP composites at different extrusion temperatures. With the increase in extrusion temperature, the bending strength and modulus of GO-CF/EP composites increase first and then decrease. The maximum bending strength and bending modulus of the composites are 728 MPa and 99.52 GPa, respectively, when the extrusion temperature is 50 °C. Compared with the extrusion temperature of 30 °C, the bending strength and bending modulus increased by 60.2% and 12.1%, respectively. The reason for this change is that the viscosity and fluidity of the matrix are greatly affected by temperature. When the extrusion temperature is low, the matrix viscosity is high, and the fluidity is poor. The process of matrix infiltration is subject to larger resistance, which affects its complete infiltration in the fiber cloth. The stress concentration is caused by the matrix distributed on the surface of each layer of the fiber cloth, and the bearing capacity of the specimen is limited due to the discontinuous structure of the internal infiltration structure. With the increase in temperature, the fluidity of the matrix is improved so that it can fill the infiltration gap, and the fiber is closely combined with the matrix. When the external force is applied, the bonded fiber effectively shares the stress, and GO dispersed in the matrix also bears and disperses the stress, which is conducive to improving the bearing capacity of the specimen. However, too high an extrusion temperature will not only increase the flow rate of the matrix, but also extrude the matrix, thus, affecting the mechanical properties of the composite.

By comparing with the existing research results, it can be seen that the bending strength of GO-CF/EP hybrid-reinforced composite prepared by VIHPS at room temperature is nearly 1.6 times higher than that of 3D needle-punched CF/epoxy resin composite, and 3.7 times higher than that of functionalized carbon nanotube-reinforced epoxy nanocomposite [29,30]. It can be seen that the mechanical properties of the micro-nano hybrid-reinforced composite prepared by GO and CF are far superior to those of CF-reinforced epoxy resin composite and nanofiller-reinforced polymer composite and have a better engineering application value.

#### 3.3.2. Fracture Microstructure and Mechanism of GO-CF/EP Composites

Figure 9a shows that the fracture surface of the fibers is disordered, and the matrix does not bond most fibers. Most fibers cannot bear the load or play a role in reinforcement. Figure 9b shows the infiltration effect and fracture mode of the composite when the extrusion temperature is 30 °C. The viscosity of the matrix is high and the fluidity is poor due to the low extrusion temperature. It is difficult to completely infiltrate the fiber gap in a limited hot-pressing time. At this extrusion temperature, most GO is distributed on the surface of the composite or between fiber layers. There is no GO distributed among the fiber bundles and less resin. In this case, the resin cannot be combined with the fiber to make the microstructure continuous, and GO cannot make the resin and fiber form a locking structure. Therefore, only a small number of fibers bear the load, and many fibers struggle to transmit stress under an external force. This causes the fibers to break at the same time. The bending strength of the composite is only 454.1 MPa. In order to obtain a compact composite with ideal mechanical properties, it is necessary to increase the extrusion temperature appropriately and analyze the relationship between the matrix distribution in the matrix and the bending properties of the composite.

By observing the fracture morphology shown in Figure 10a, it can be found that when the extrusion temperature is 50 °C, fiber bundles are closely bonded by the matrix, and the fracture surface is uneven. There are holes after the fiber extraction. When the load of some CF reaches the fatigue load, it fractures and transfers the load to other fibers to enhance the mechanical properties of the composite. Figure 10b shows that there are uniformly distributed matrices in the fiber layers and fiber gap of the composite. The existence of GO causes the resin and CF closely combine and enhances the interface bonding energy between the matrix and the reinforcement. When an external force is applied to the composite, the stress is transferred from one part of the fiber to the other through the matrix. The flow ability of the matrix is enhanced at 50 °C, and the fiber gap is filled under the effect of the extrusion force, which improves the density of the composite. The bending strength of the specimens prepared at this extrusion temperature is 728 MPa, which indicates that the mechanical properties of the composites have been greatly improved. From the double Arrhenius viscosity model shown in Equation (4), the resin viscosity decreases with the increase in temperature [31]. The gradual decrease in viscosity promotes the increase in resin fluidity, so that the matrix is fully immersed in the fiber bundle and the filament is bonded together. Therefore, composites have good bearing capacity.
(4)$$\ln\eta (T,t)={\mathrm{lnZ}}_{\infty }+\frac{E_{Z}}{RT}+K_{0}e^{-\frac{E_{T}}{RT}}t$$
where η is the viscosity of matrix, Z_∞_ and K_0_ are the Arrhenius pre-exponential factors, $E_{z}$ is the activation energy of the solution flow, $E_{t}$ is the activation energy of curing reaction, *R* is the gas molar constant, and *T* and *t* are reaction temperature and reaction time, respectively.

Figure 11a shows that the matrix distribution at the fracture is highly uneven, and the number of cracks increases significantly. The fracture plane of the fiber is neat, showing typical brittle fracture characteristics, indicating that the fiber does not have a good bearing capacity. When the extrusion temperature is 70 °C, the ethanol solution used to disperse GO volatilizes, resulting in some defects such as holes or small cracks in the composite. These defects gradually expand along the direction of weak interface bonding, which leads to fracture or the delamination failure of the composite. Figure 11b shows the infiltration effect, fracture mode and stress transfer mode of the composites when the extrusion temperature is 70 °C. When the composite is at this extrusion temperature, the matrix is partially extruded out due to a remarkable improvement of fluidity, which directly leads to the appearance of less matrix area in the composite. The area with less resin content is mainly an elliptical hole parallel to the fiber direction [32]. The appearance of these areas is a direct factor in the existence of cracks. However, it is not difficult to find that at this extrusion temperature, the crack is narrow and bent because the existence of GO deflects the crack direction, effectively reducing the stress concentration. The bending strength of the composite is 401.6 MPa. With the increase in extrusion temperature, the fluidity of the matrix is better, but part of the matrix is extruded, and the phenomenon of uneven infiltration appears. The generation of bubbles causes the internal structure of the composite to show defects, which seriously affects the performance of the specimen.

## 4. Conclusions

(1) GO-CF/EP composites were prepared by the VIHPS method. When the extrusion temperature is 50 °C, the surface morphology of the composite is ideal, and no obvious defects such as holes and warps are found.

(2) The addition of GO increases the contact area between EP and CF, making them form a mechanical lock. The appearance of this structure significantly improves the interface bonding ability of the composite. When the composite is subjected to external force, GO bears part of the stress, which reduces the stress concentration in the crack area, deflects the crack, and significantly improves the mechanical properties of the composite.

(3) When the extrusion temperature was 50 °C, the matrix was evenly distributed between CF. GO not only bonded the EP and CF together by mechanical locking, but also effectively shared the load. Due to the bonding effect of resin, the fibers could give full play to their load-bearing capacity, and most of the fibers were pulled out or broken when they broke. The bending strength at this extrusion temperature was 728 MPa.

(4) It was found that too high or too low an extrusion temperature would affect the internal structure and mechanical properties of the composites. When the extrusion temperature is low, the viscosity of the matrix is high and the flow rate is slow, leading to the agglomeration and solidification of the matrix. Most of the matrix and fiber cannot fully play their respective roles. In this case, the bending strength of the composite is low. When the extrusion temperature is high because the matrix has good fluidity, it is partially extruded, and the ability of the matrix to transfer load is weakened. The ability of GO to share stress is also weakened, resulting in the poor mechanical properties of the composites. Therefore, a reasonable selection of extrusion temperature in the process of composite molding is the key to preparing excellent composites.

## Figures and Tables

**Figure 1 nanomaterials-12-03839-f001:**
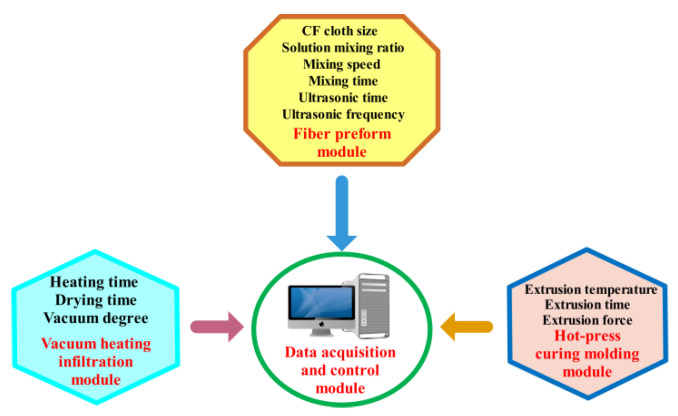
Vacuum infiltration hot-pressing experimental system.

**Figure 2 nanomaterials-12-03839-f002:**
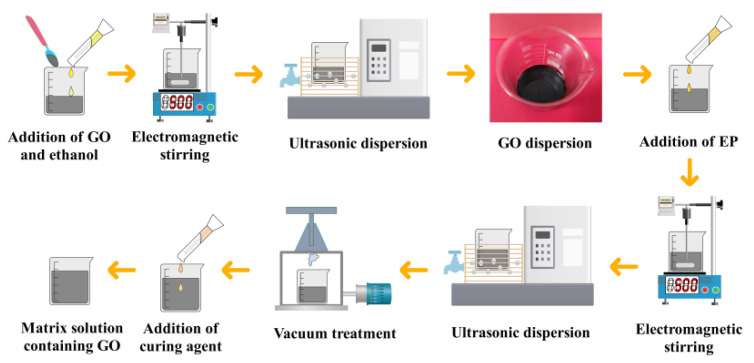
Preparation of curing mixed solution.

**Figure 3 nanomaterials-12-03839-f003:**
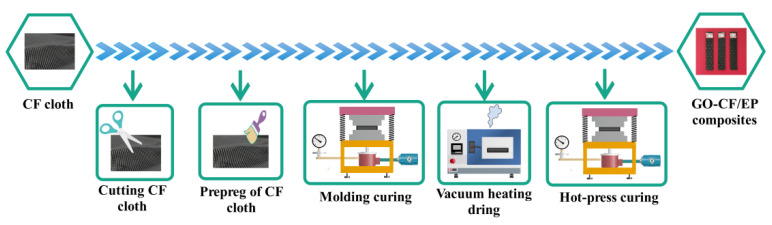
Preparation process of GO-CF/EP composites.

**Figure 4 nanomaterials-12-03839-f004:**
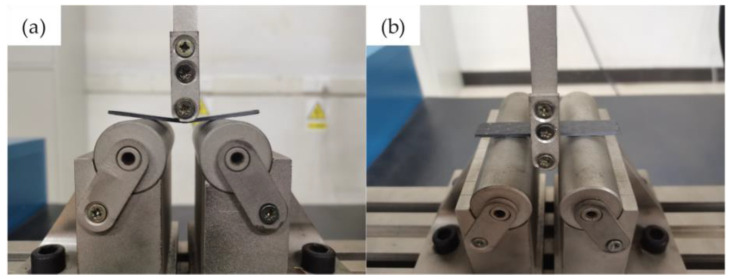
(**a**) Front view of composite specimen under three-point bending load; (**b**) Top view of composite specimen under three-point bending load.

**Figure 5 nanomaterials-12-03839-f005:**
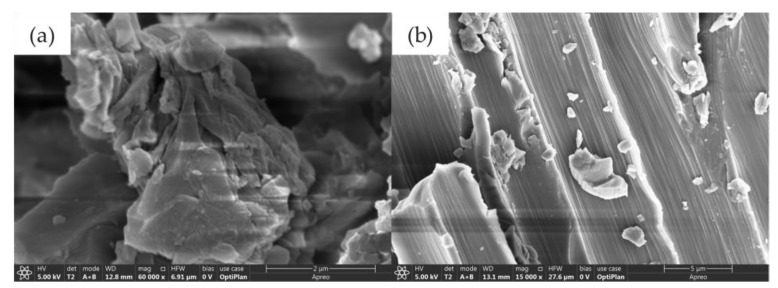
The SEM of GO (**a**) GO/EP composite (60,000×), (**b**) GO-CF/EP composite (15,000×).

**Figure 6 nanomaterials-12-03839-f006:**
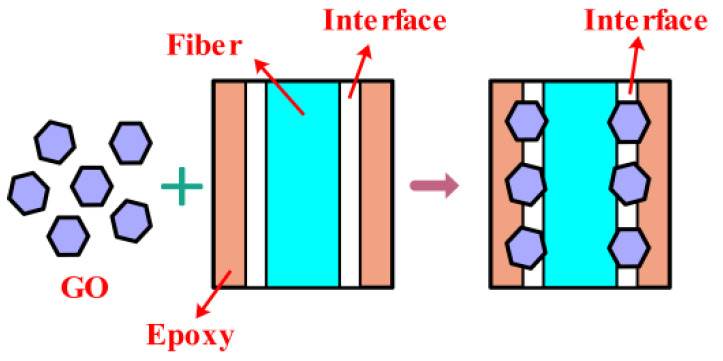
Schematic diagram of mechanical interlocking.

**Figure 7 nanomaterials-12-03839-f007:**
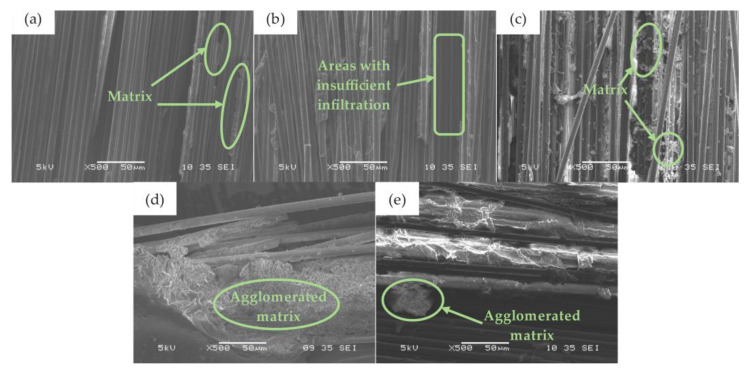
Infiltration structure of GO-CF/EP composites at different extrusion temperatures: (**a**) 30 °C, (**b**) 40 °C, (**c**) 50 °C, (**d**) 60 °C, (**e**) 70 °C.

**Figure 8 nanomaterials-12-03839-f008:**
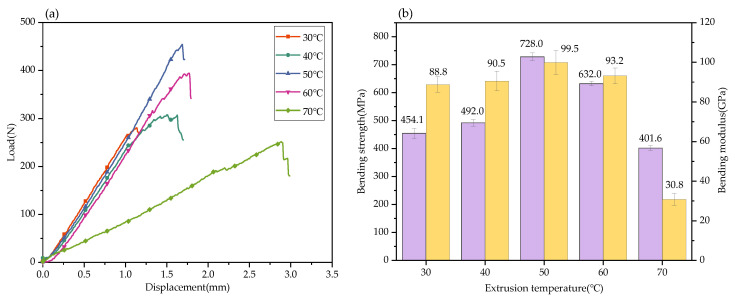
Three-point bending test results of GO-CF/EP composites (**a**) load–displacement curve, (**b**) bending strength and bending modulus.

**Figure 9 nanomaterials-12-03839-f009:**
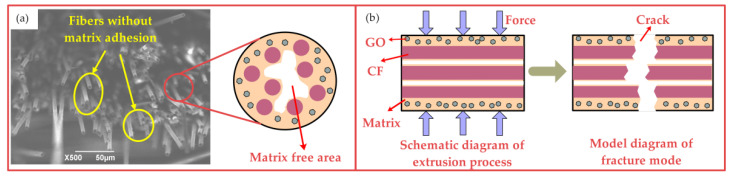
Fracture microstructure and mechanism of GO-CF/EP composite (30 °C): (**a**) fracture microstructure, (**b**) fracture mechanism.

**Figure 10 nanomaterials-12-03839-f010:**
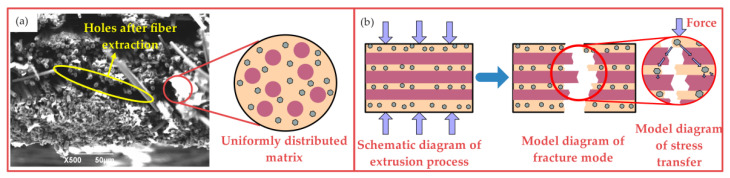
Fracture microstructure and mechanism of GO-CF/EP composite (50 °C): (**a**) fracture microstructure, (**b**) fracture mechanism.

**Figure 11 nanomaterials-12-03839-f011:**
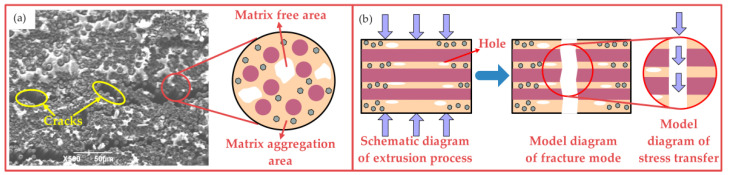
Fracture microstructure and mechanism of GO-CF/EP composite (70 °C): (**a**) fracture microstructure, (**b**) fracture mechanism.

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
