# Peer review of "Influence of Extrusion Temperature on Properties of Graphene Oxide-Carbon Fiber/Epoxy Composite Prepared by Vacuum Infiltration Hot-Press-Forming Experimental System"

_nanomaterials, 2022, doi:10.3390/nano12213839_

Round 1

Reviewer 1 Report

This is an interesting manuscript reporting on the temperature dependence of the microstructure of epoxy composites cotanining a micture of filler particles (carbon microfibers and graphene oxide) obtained by a speciific extrusion process.

I think it is novel and well written enough to be publishable in Nanomaterials. The aims of the work is described clearly, and the novelty is that effect of extrusion temperature on the infiltration structure 78 and mechanical properties of GO-CF/EP composites have been studied systematically for the first time.

However, I have the following comments that need to be clarified before publication.

1) the title ends as "Prepared by VIHPS"...please write out the full name of the process, because acronyms should not be used in titles.

2) first sentence of introduction: also here, write out the acronym: CFRP. It should be defined at fisrt mention.

3) line 35: "the exfoliation of original graphene": graphene is by definition an elementary sheet of graphene, so it cannot be exfoliated. Correct phrase is "the exfoliation of graphite"

4) line 41: "In addition, some chemical functional groups on the surface of GO combine with the matrix to make them fully contact": it does not seem very scientific, do you mean the the chemical functiona groups of GO form covalent bonds with the polymeric matrix and this makes the structure stronger? If a filler is inside the matrix, I think it means that it is "fully contacted" because it is entirely surrounded by the matrix polymer. 

5) "At present, researchers at home and abroad have conducted many experiments": this reads strange here, what do the authors mean about "home and abroad"? In international science publications it is usually not relevant to mention in which countries were experiments conducted. If there is any special importance why is important to mention research in China vs. outside China, please provide the reason.

6) "using an improved Hummer process": the correct name is: Hummers(-Offeman)

7) " Its sheet diameter is 0.2-10 μm, the thickness is 1nm, the number of layers is 1-2 and the purity is 99%.": are these data provided by the manufacturer or confirmed in the author's lab? The expression: "number of layers" should be modified: readers can think that there are alltogether 1-2 layers in the sample, while the authors must mean the average number of stacked GO sheets in dispersion.

8) The sentence :"In general, because GO is insoluble in the matrix, there are molecular forces between the matrix and GO,"is not very reasonable because GO is not insoluble becasuse there are molecular forces, in general. Molecular forces are dominantly attractive in all materials which are liquids and solids (otherwise they would be gases), but "solubility" depends on the balance of attractive and repulsive molecular interatcions.

9) What does "vacuum of -0.09 MPa" mean? Negative pressures make no sense, so I think here the authors mean a relative pressure or pressure difference. Please explain/specify!

10) There are 20 cited references, and they are fine. However, I observe that all of them are from the 2010's....while there were GO/polymer nanocomposites published already 30 years ago. Maybe they are not difectly relevant to this study, but "graphite oxide"/polymer papers were known even before the scothch-tape isolation of "graphene"!

I suggest that the authors check these 1990-2000's papers, mostly in the journal "Carbon" and if they are useful, cite a few.

Author Response

Dear reviewer:

Thank you very much for your kindly comments of our manuscript entitled “Influence of Extrusion Temperature on Properties of Graphene Oxide-Carbon Fiber/Epoxy Composite Prepared by VIHPS” (Manuscript ID: nanomaterials - 1948550). Your comments are very valuable for improving my paper, and replies to your comments are attached. Revisions in the manuscript are marked in red, and these changes will not influence the main content and framework of the paper. Meanwhile, we also have revised the whole manuscript carefully and improved the language expressions.

The main corrections in the manuscript and the responses to your comments are listed point by point in the next page.

Correspondence and phone calls about this paper should be directed to Prof. Yuqin Ma at the following address, phone and e-mail:

Corresponding Author:

Prof. Yuqin Ma

Key Laboratory of Road Construction Technology and Equipment of MOE

School of Construction Machinery

Chang'an University

No.126, Middle Section of Erhuan South Road, Xi'an, Shannxi 710064, China

E-mail: [email protected] (Y.M.)

With kind personal regards,

Sincerely yours,

Prof. Yuqin Ma

2022-10-14

Remarks:

Reviewer #1:

This is an interesting manuscript reporting on the temperature dependence of the microstructure of epoxy composites cotanining a micture of filler particles (carbon microfibers and graphene oxide) obtained by a speciific extrusion process.

I think it is novel and well written enough to be publishable in Nanomaterials. The aims of the work is described clearly, and the novelty is that effect of extrusion temperature on the infiltration structure 78 and mechanical properties of GO-CF/EP composites have been studied systematically for the first time.

However, I have the following comments that need to be clarified before publication.

Q1: The title ends as "Prepared by VIHPS"...please write out the full name of the process, because acronyms should not be used in titles.

R1: Thank you very much for your valuable opinions on the manuscript. According to the suggestion, we have revised the title of the paper from "Influence of Extrusion Temperature on Properties of Graphene Oxide-Carbon Fiber/Epoxy Composite Prepared by VIHPS" to "Influence of Extrusion Temperature on Properties of Graphene Oxide-Carbon Fiber/Epoxy Composite Prepared by Vacuum Infiltration Hot-Press Forming Experimental System". The title is more comprehensive and specific after being revised according to your opinion.

Q2: First sentence of introduction: also here, write out the acronym: CFRP. It should be defined at fisrt mention.

R2: Thank you for your preciseness and carefulness. Due to our carelessness, the definition of CFRP was not presented in the manuscript. I am sorry for the inconvenience caused to your review work. According to your suggestion, we have explained in detail the full name of CFRP: carbon fiber reinforced polymer in the appropriate place of the article. In order to avoid the recurrence of such problems, we have carefully checked and modified all the contents of the manuscript. Thank you for your question.

Q3: Line 35: "the exfoliation of original graphene": graphene is by definition an elementary sheet of graphene, so it cannot be exfoliated. Correct phrase is "the exfoliation of graphite"

R3: Thank you for correcting our mistakes. After reading the relevant literature, as you said, GO is the exfoliation of graphene during oxidation. We have modified this part. We can see that you are rigorous scientific research, which is worth learning from you.  

Q4: Line 41: "In addition, some chemical functional groups on the surface of GO combine with the matrix to make them fully contact": it does not seem very scientific, do you mean the the chemical functiona groups of GO form covalent bonds with the polymeric matrix and this makes the structure stronger? If a filler is inside the matrix, I think it means that it is "fully contacted" because it is entirely surrounded by the matrix polymer.

R4: Thank you very much for your question. Through further thinking about your question and searching for relevant literature, the correct expression of this sentence should be: "In addition, the chemical functional groups on the GO surface react with the groups on the fiber surface, which promotes wettability between matrix and CF, and enhances the interface adhesion between them, so the composites have better interface performance [14]." As you said, GO is dispersed in the matrix and surrounded by the matrix. It can not only toughen the matrix but also disperse the crack transmission path in the matrix and enhance the bearing capacity of the composite. Thanks again for your question.

14.Li, M.; Gu, Y.Z.; Liu, Y.N.; Li, Y.X.; Zhang, Z.G. Interfacial improvement of carbon fiber/epoxy composites using a simple process for depositing commercially functionalized carbon nanotubes on the fibers. Carbon 2013,52:109-121.

Q5: "At present, researchers at home and abroad have conducted many experiments": this reads strange here, what do the authors mean about "home and abroad"? In international science publications it is usually not relevant to mention in which countries were experiments conducted. If there is any special importance why is important to mention research in China vs. outside China, please provide the reason.

R5: Thank you very much for your guidance. We have revised this sentence. Both research in China and outside China are equally important because all research results can show the development status of this field only when combined. I am sorry for the misunderstanding caused by our writing mistakes. Thanks again for your preciseness. We have learned a lot from these problems to avoid similar mistakes in the future.

Q6: "using an improved Hummer process": the correct name is: Hummers(-Offeman)

R6: Thank you for correcting our mistakes. We have changed "Hummer" to "Hummers" at the corresponding position in the original text. At the same time, thank you for expanding our knowledge about the GO preparation method proposed by Hummers and Offeman in 1958.

Q7: " Its sheet diameter is 0.2-10 μm, the thickness is 1nm, the number of layers is 1-2 and the purity is 99%.": are these data provided by the manufacturer or confirmed in the author's lab? The expression: "number of layers" should be modified: readers can think that there are alltogether 1-2 layers in the sample, while the authors must mean the average number of stacked GO sheets in dispersion.

R7: Thank you very much for your question. Various performance parameters mentioned in the Experimental materials were provided by the manufacturer. In this part, the number of GO layers refers to the number of layers of GO powder, including single layer and double layer. The surface and boundary of each carbon atom layer are connected with oxygen-containing functional groups. However, graphene oxide presents a slight agglomeration state in the composite matrix in this paper, thus forming a multi-layer structure. In order to reduce the agglomeration degree of GO, an electromagnetic stirring device and ultrasonic dispersion device have been used to fully disperse the solidified mixed solution containing GO. In contrast, GO still has some agglomeration phenomena. The complete and uniform dispersion of GO in polymer matrix has always been a research hotspot in related fields. Thank you for your guidance. In the future scientific research work, we will do a more in-depth discussion on this issue. In addition, we changed the "number of layers is 1-2" to "layer number of carbon atoms connected with oxygen-containing functional groups of GO powder of 1-2". Thank you again for your question.

Q8: The sentence:"In general, because GO is insoluble in the matrix, there are molecular forces between the matrix and GO,"is not very reasonable because GO is not insoluble becasuse there are molecular forces, in general. Molecular forces are dominantly attractive in all materials which are liquids and solids (otherwise they would be gases), but "solubility" depends on the balance of attractive and repulsive molecular interatcions.

R8: Thank you for your question. As you said, GO is not insoluble. Molecular force exists objectively, and the solubility of GO depends on the interaction of gravitation and repulsion. However, its two-dimensional surface structure, high surface energy, strong hydrogen bond, dipole interaction of interlayer oxygen-containing groups, dispersion, and stripping of GO in the resin matrix are hindered [12]. For this reason, the experimental process in this paper uses the wet transfer method to disperse GO in an ethanol solution and then further disperse the GO suspension through the cavitation effect of ultrasonic dispersion, which can effectively reduce the aggregation of particulate GO in the matrix solution. Based on the above analysis, we re-modify this sentence: "Due to the two-dimensional surface structure, high surface energy, strong hydrogen bond and dipole interaction of interlayer oxygen-containing groups of GO, the dispersion and shedding behavior of GO in the resin are hindered."

12.Ren, J.J.; Chen, L.; Liu, Z.Q.; Song, Q.H.; Liu, C.Z. Study on the heat transfer reinforcement of glass fiber/epoxy resin composites by grafting and dispersing graphene oxide. Composites Science and Technology 2021,216:109039.

Q9: What does "vacuum of -0.09 MPa" mean? Negative pressures make no sense, so I think here the authors mean a relative pressure or pressure difference. Please explain/specify!

R9: We are sorry that our writing has misled you. -0.09 MPa is the data displayed by the vacuum gauge. The data is the relative pressure measured based on the atmospheric pressure. The standard atmospheric pressure is 100 KPa. From this relative pressure, it can be seen that the absolute pressure is less than the atmospheric pressure. Therefore, the absolute pressure in the vacuum pump is less than one atmospheric pressure, so that it can be interpreted as the vacuum degree is -0.09 MPa. In order to avoid misunderstanding, we have revised this sentence. Thank you again for your preciseness and carefulness.

Q10: There are 20 cited references, and they are fine. However, I observe that all of them are from the 2010's....while there were GO/polymer nanocomposites published already 30 years ago. Maybe they are not difectly relevant to this study, but "graphite oxide"/polymer papers were known even before the scothch-tape isolation of "graphene"!

I suggest that the authors check these 1990-2000's papers, mostly in the journal "Carbon" and if they are useful, cite a few.

R10: Thank you very much for your valuable opinions on the paper. According to your suggestion, we searched the articles about GO/polymer nanocomposites published on "Carbon" from 1990 to 2000 and after 2000. At the same time, we also found such articles published in other journals during this period and discussed and cited them in the introduction. The specific references are as follows:

The study of GO-reinforced polymer matrix was earlier than that of GO-reinforced CFRP as nanofiller. Xiao and others [9] proposed a new method to synthesize polyaniline-intercalated GO nanocomposites. The experimental results showed that the compound had a higher crystallinity regularity. Szabo et al. [10] deposited thin GO/cationic polymer (poly(diallyldimethylammoniumchloride, PDDA) multilayer films on a glass substrate through wet chemical self-assembly. Through experimental research, they found that the multilayer combination of GO and PDDA reduced to a non-conductive turbine layer carbon nanofilm on the glass. Matsuo et al. [11] prepared poly(epoxy oxide)-intercalated graphite oxides (PEO-intercalated GOs) with different c-axis repeat distances and characterized them in detail. The decomposition temperature of GO was different, and GO decomposed at higher temperatures was accompanied by the destruction of the carbon skeleton.

  • Xiao, P.; Xiao, M.; Liu, P.G.; Gong, K.C. Direct synthesis of a polyaniline - intercalated graphite oxide nanocomposite. Carbon 2000,38(4):626-628.
  • Szabo, T.; Szeri, A.; Dekany, I. Composite graphitic nanolayers prepared by self-assembly between finely dispersed graphite oxide and a cationic polymer. Carbon 2005,43(1):87-94.
  • Matsuo,Y.;Tahara, K.; Sugie, Structure and thermal properties of poly (ethylene oxide)-intercalated graphite oxide. Carbon 1997,35(1): 113-120.

Reviewer 2 Report

the manuscript show  the study about the effects of extrusion temperature on the microstructure, fracture mech- 16 anism and mechanical properties of GO-CF/EP composites.  It was found that the best mechanical property of composites and infiltration effect of the matrix in the fiber gap were obtained at the temperature of 50℃ and the bending strength of the composite reached 728 MPa.

The manuscript is clearly and original. the experimantal results and the discussion are supported by adapt references, for this it can be pubblished in present form.

Author Response

Dear reviewer:

Thank you very much for your kindly comments of our manuscript entitled “Influence of Extrusion Temperature on Properties of Graphene Oxide-Carbon Fiber/Epoxy Composite Prepared by VIHPS” (Manuscript ID: nanomaterials - 1948550). Your comments are very valuable for improving my paper, and replies to your comments are attached. Revisions in the manuscript are marked in red, and these changes will not influence the main content and framework of the paper. Meanwhile, we also have revised the whole manuscript carefully and improved the language expressions.

The main corrections in the manuscript and the responses to your comments are listed point by point in the next page.

Correspondence and phone calls about this paper should be directed to Prof. Yuqin Ma at the following address, phone and e-mail:

Corresponding Author:

Prof. Yuqin Ma

Key Laboratory of Road Construction Technology and Equipment of MOE

School of Construction Machinery

Chang'an University

No.126, Middle Section of Erhuan South Road, Xi'an, Shannxi 710064, China

E-mail: [email protected] (Y.M.)

With kind personal regards,

Sincerely yours,

Prof. Yuqin Ma

2022-10-17

Remarks:

Reviewer #2:

The manuscript show the study about the effects of extrusion temperature on the microstructure, fracture mech- 16 anism and mechanical properties of GO-CF/EP composites. It was found that the best mechanical property of composites and infiltration effect of the matrix in the fiber gap were obtained at the temperature of 50℃ and the bending strength of the composite reached 728 MPa.

The manuscript is clearly and original. the experimantal results and the discussion are supported by adapt references, for this it can be pubblished in present form.

R: Thank you very much for your comments on the paper and for your time and effort in reviewing the paper. It is my honor to receive your positive comments on this manuscript. In future scientific research work, we will continue to work hard, and thank you again for your decision.

Reviewer 3 Report

The paper with title "Influence of Extrusion Temperature on Properties of Graphene Oxide-Carbon Fiber/Epoxy Composite Prepared by VIHPS" evaluates the effect of extrusion temperature on the infiltration structure and mechanical properties of GO-CF/EP composites, the micro-nano scale hybrid reinforced composites were prepared by VIHPS in this study. Results reported are clears and are of potential interest to readers of the Nanomaterials journal and worthy of publication, but some suggestions among the text are reported in the MS. Furthermore, I recommend a minor revision of the MS.

Author Response

Dear reviewer:

Thank you very much for your kindly comments of our manuscript entitled “Influence of Extrusion Temperature on Properties of Graphene Oxide-Carbon Fiber/Epoxy Composite Prepared by VIHPS” (Manuscript ID: nanomaterials - 1948550). Your comments are very valuable for improving my paper, and replies to your comments are attached. Revisions in the manuscript are marked in red, and these changes will not influence the main content and framework of the paper. Meanwhile, we also have revised the whole manuscript carefully and improved the language expressions.

The main corrections in the manuscript and the responses to your comments are listed point by point in the next page.

Correspondence and phone calls about this paper should be directed to Prof. Yuqin Ma at the following address, phone and e-mail:

Corresponding Author:

Prof. Yuqin Ma

Key Laboratory of Road Construction Technology and Equipment of MOE

School of Construction Machinery

Chang'an University

No.126, Middle Section of Erhuan South Road, Xi'an, Shannxi 710064, China

E-mail: [email protected] (Y.M.)

With kind personal regards,

Sincerely yours,

Prof. Yuqin Ma

2022-10-17

Remarks:

Reviewer #3:

The paper with title "Influence of Extrusion Temperature on Properties of Graphene Oxide-Carbon Fiber/Epoxy Composite Prepared by VIHPS" evaluates the effect of extrusion temperature on the infiltration structure and mechanical properties of GO-CF/EP composites, the micro-nano scale hybrid reinforced composites were prepared by VIHPS in this study. Results reported are clears and are of potential interest to readers of the Nanomaterials journal and worthy of publication, but some suggestions among the text are reported in the MS. Furthermore, I recommend a minor revision of the MS.

Q1: Line 15: you could write: 10, 40, 50, 60 and 70 ºC.

R1: Thank you for your preciseness and carefulness. According to your suggestion, we have revised this sentence. In order to avoid similar errors, we have carried out a detailed inspection and modification of the full text.

Q2: Line 38: you could write: It.

R2: Thank you for your suggestion. We have changed "GO" to "It".

Q3: Line 60: SiCp: you could write the full nameï¼›Line 63: SiCp/Mg-Zn-Ca: you could write the full nameï¼›Line 69: 2D-Cf/Al: you could write the full name.

R3: Thank you very much for your preciseness. We have appropriately added the complete form in the corresponding references to the manuscript. After revising according to your suggestions, make the article more complete and specific. The full name of "SiCp" is "silicon carbide particles", the full name of "Mg-Zn-Ca" is "Mg-Zn-Ca matrix" and "2D-Cf/Al composites" is "two-dimensional carbon fiber reinforced aluminum matrix composites”.

Reviewer 4 Report

This article deals with constructing composite materials and mechanical properties. Interesting topic, however, before considering it need clarification and revision base on the following comments

Technical issues:

1. Expand the abbreviation at first used in the text. For example Line31, CFRP.

2. Give enough space between numerical values and their units, for example, Line 15, 30 oC, etc.

Scientific issues

3. Introduction should be improved by discussing the significance and novelty of this study compared with existing literature.

4. The authors should be discussed the details of characterization instruments and sample preparation for each characterization analysis.

5. The novelty and significance of this study should be explored in the results and discussion by comparing other reports.  Their response indicates each line, where discussed novelty and significance of these results compared other.

Author Response

Dear reviewer:

Thank you very much for your kindly comments of our manuscript entitled “Influence of Extrusion Temperature on Properties of Graphene Oxide-Carbon Fiber/Epoxy Composite Prepared by VIHPS” (Manuscript ID: nanomaterials - 1948550). Your comments are very valuable for improving my paper, and replies to your comments are attached. Revisions in the manuscript are marked in red, and these changes will not influence the main content and framework of the paper. Meanwhile, we also have revised the whole manuscript carefully and improved the language expressions.

The main corrections in the manuscript and the responses to your comments are listed point by point in the next page.

Correspondence and phone calls about this paper should be directed to Prof. Yuqin Ma at the following address, phone and e-mail:

Corresponding Author:

Prof. Yuqin Ma

Key Laboratory of Road Construction Technology and Equipment of MOE

School of Construction Machinery

Chang'an University

No.126, Middle Section of Erhuan South Road, Xi'an, Shannxi 710064, China

E-mail: [email protected] (Y.M.)

With kind personal regards,

Sincerely yours,

Prof. Yuqin Ma

2022-10-17

Remarks:

Reviewer #4:

This article deals with constructing composite materials and mechanical properties. Interesting topic, however, before considering it need clarification and revision base on the following comments.

Q1: Expand the abbreviation at first used in the text. For example Line31, CFRP.

R1: Thank you for your preciseness and carefulness. Due to our carelessness, the definition of CFRP was not presented in the manuscript. I am sorry for the inconvenience caused to your review work. According to your suggestion, we have explained in detail the full name of CFRP: carbon fiber reinforced polymer in the appropriate place of the article. In order to avoid the recurrence of such problems, we have carefully checked and modified all the contents of the manuscript. Thank you for your question.

Q2: Give enough space between numerical values and their units, for example, Line 15, 30 â„ƒ, etc.

R2: Thank you very much for your guidance on the manuscript. According to your suggestion, we have added spaces between values and units. At the same time, we recheck the full text to prevent similar errors.

Q3: Introduction should be improved by discussing the significance and novelty of this study compared with existing literature.

R3: Thank you very much for your valuable opinions on the paper. In the Introduction, we have discussed the importance and innovation of this study in combination with existing references. Specifically, it includes: (1) aiming at the problems of nanomaterials/polymer composites and CF reinforced resin matrix composites, this paper prepared hybrid reinforced composites with GO and CF as reinforcements and epoxy resin as the matrix. The strengthening effect of GO on the mechanical properties of composites was studied from the bonding effect of GO on the reinforcement/matrix interface, the force transfer effect on the specimen, and the defect control. (2) In this paper, GO-CF/EP composites were prepared by the self-developed vacuum infiltration hot-press forming experimental system (VIHPS), which can reduce the preparation time and improve the preparation efficiency. At the same time, the real-time control of various process parameters could be realized to ensure that the composite had an ideal penetration effect and effectively reduced the probability of defects. (3) As an essential process parameter in the hot pressing curing process, the extrusion temperature had a significant influence on the fluidity of the matrix, the infiltration effect, and the defect control of the composite. However, the existing research could not solve the above problems. Therefore, composites with different extrusion temperatures were prepared in this paper. By discussing the infiltration structure, fracture morphology, and mechanical properties, the influence of extrusion temperature on the infiltration and curing of composites was determined to clarify the strengthening rule of mechanical properties.

Q4: The authors should be discussed the details of characterization instruments and sample preparation for each characterization analysis.

R4: Thank you for your comments. In the Testing and characterization section, we have introduced in detail instruments and the preparation of test pieces for the three-point bending test and microstructure test.

The three-point bending test instrument is the DNS 100 electronic universal testing machine of Changchun machinery institute, which applies to bending, tensile, compression, and other mechanical property tests. It can calculate the maximum test force, bending strength, tensile strength, compressive strength, elastic modulus, and other parameters. The maximum test force can provide 100 kN, and the measurement accuracy is ±0.5%. According to GB/T3356-2014, the size of the specimen used for the three-point bending test is 80 mm×12.5 mm×2 mm. Composites with the same extrusion temperature need to test 5 test pieces. Therefore, the size prepared in 2.2.2 is 100 mm×95 mm×2 mm composite is cut into 6 pieces with a size of 80 mm×12.5 mm×2 mm test piece, five of which are used for bending property test and one for cutting sample for microstructure test.

The instrument used for microstructure testing and fracture morphology characterization is the JEOL JSM-6390A scanning electron microscope, a high vacuum analytical scanning electron microscope purchased from Japan Electronics Co., Ltd. The size of the specimen for microstructure testing is 10 mm×12.5 mm×2 mm, the specimen for fracture morphology test needs to be cut after a three-point bending test of the specimen, and the size is close to 10 mm×12.5 mm×2 mm. Three specimens are tested at the same extrusion temperature.

Thank you again for your suggestions. The revised article is clearer and more complete.

Q5: The novelty and significance of this study should be explored in the results and discussion by comparing other reports. Their response indicates each line, where discussed novelty and significance of these results compared other.

R5: Thank you very much for your suggestions on the Results and discussion. We searched some related papers. Through comparison, it was found that the bending properties of the micro-nano scale hybrid reinforced composites prepared in this paper were 1.6 times higher than those of CF reinforced epoxy composites and 3.7 times higher than those of functionalized carbon nanotube-reinforced epoxy nanocomposites. We have added these contents to the Bending properties of the GO-CF/EP composites section. The supplemented test conclusions can verify the innovation of this paper and the excellent performance of the composites prepared. Thank you again for your guidance.

Round 2

Reviewer 4 Report

The revised form is given sufficient information, the present form can be accepted.